# Immunogenetic Predictors of Severe COVID-19

**DOI:** 10.3390/vaccines9030211

**Published:** 2021-03-03

**Authors:** Anna Malkova, Dmitriy Kudlay, Igor Kudryavtsev, Anna Starshinova, Piotr Yablonskiy, Yehuda Shoenfeld

**Affiliations:** 1St. Petersburg State University, Saint Petersburg 199034, Russia; piotr_yablonskii@mail.ru (P.Y.); yehuda.shoenfeld@sheba.health.gov.il (Y.S.); 2Sechenov First Moscow State Medical University, Moscow 119435, Russia; D624254@gmail.com; 3NRC Institute of Immunology FMBA of Russia, Moscow 115478, Russia; 4FSBI Institute of Experimental Medicine, St. Petersburg 197376, Russia; igorek1981@yandex.ru; 5Far Eastern Federal University, Vladivostok 690091, Russia; 6FSBI V.A. Almazov National Medical Research Center, Ministry of Health of Russia, St. Petersburg 197241, Russia; starshinova_777@mail.ru; 7St. Petersburg Research Institute of Phthisiopulmonology, Saint Petersburg 191036, Russia; 8Zabludowicz Center for Autoimmune Diseases, Sheba Medical Center, Tel-Hashomer 5265601, Israel; 9Sackler Faculty of Medicine, Tel-Aviv University, Tel-Aviv 6997801, Israel

**Keywords:** new coronavirus infection, COVID-19, biomarkers, immunological parameters, genes, HLA, ACE-2

## Abstract

According to an analysis of published data, only 20% of patients with the new coronavirus infection develop severe life-threatening complications. Currently, there are no known biomarkers, the determination of which before the onset of the disease would allow assessing the likelihood of its severe course. The purpose of this literature review was to analyze possible genetic factors characterizing the immune response to the new coronavirus infection that could be associated with the expression of angiotension-converting enzyme 2 (ACE-2) and related proteins as predictors of severe Corona virus disease 2019 (COVID-19). We analyzed original articles published in Medline, PubMed and Scopus databases from December 2019 to November 2020. For searching articles, we used the following keywords: New coronavirus infection, Severe acute respiratory syndrome-related coronavirus 2 (SARS-CoV-2), COVID-19, severe course, complications, thrombosis, cytokine storm, ACE-2, biomarkers. In total, 3714 publications were selected using the keywords, of which 8 were in congruence with all the criteria. The literature analysis of the association of immunogenic characteristics and the expression of ACE-2 and related proteins with the development of severe COVID-19 revealed following genetic factors: HLA-B*46:01 genotype, CXCR6 gene hypoexpression, CCR9 gene expression, TLR7, rs150892504 mutations in the ERAP2 gene, overexpression of wild-type ACE-2, TMPRSS2 and its different polymorphisms. Genes, associated with the severe course, are more common among men. According to the analysis data, it can be assumed that there are population differences. However, the diagnostic significance of the markers described must be confirmed with additional clinical studies.

## 1. Introduction

An outbreak of the new coronavirus infection (COVID-19) was first identified in early December 2019 in the city of Wuhan and caused a global pandemic. From December 2019 to November 2020, the number of COVID-19 infected patients worldwide has been more than 50 million (https://www.worldometers.info/coronavirus/) (accessed on 11 November 2020). The disease manifests as acute respiratory infection characterized by fever, cough and shortness of breath in the majority of patients; some develop neurological symptoms such as hypo-/anosmia and dysgeusia [1,2]. Unfortunately, around 20% of coronavirus-infected patients progress to the development of viral pneumonia, while some of them are likely to develop a very severe form of the disease. The life-threatening complications of the disease include cardiovascular disorders and systemic inflammatory response, which leads to the development of acute respiratory distress syndrome (ARDS) [3].

A severe course of the coronavirus infection develops in 14% of cases and is characterized by fever with severe dyspnea, respiratory failure, tachypnea (>30 breaths per min) and hypoxia (oxygen saturation SpO2 < 90% indoors). ARDS can develop in 5% of patients. The syndrome indicates serious, newly developing respiratory failure or worsening of pre-existing respiratory symptoms [4].

Unrestricted inflammatory infiltration of immune cells is observed in the lungs, which, in addition to a direct viral affection, contributes to a greater damaged tissue due to the excessive secretion of proteases and reactive oxygen species. Diffuse damaged alveolar is observed, characterized by desquamation of alveolar cells, formation of hyaline membranes, development of pulmonary edema and pulmonary fibrosis [4,5,6]. Due to disturbances in the regulation of the cardiovascular system, extensive thrombus formation, ventilation and perfusion deterioration, dysregulation of blood vessels develops in response to hypoxia. High D-dimers and lactate dehydrogenase (LDH) levels can be detected in the peripheral blood. Microcirculation disorders can be observed in the kidneys, brain and other vital organs in the late stages of acute respiratory distress syndrome [6].

An analysis of the pathogenetic aspects of the development of severe forms of COVID-19 showed that immunocompetent cells and ACE-2-expressing cells are the most important elements involved in the described processes. According to some studies, there is a relationship between the ACE-2 structure in the male population and various populations and different severity of the COVID-19 course.

In this regard, we put forward a review exploring the existence of genetic factors characterizing the immune response in the new coronavirus infection, as well as genes associated with the expression of ACE-2 and related proteins. The study’s purpose is to analyze the literature to determine possible biomarkers for the prediction of severe COVID-19 outcome.

## 2. Design of the Study 

We studied original articles published in Medline, PubMed and Scopus databases from December 2019 to November 2020.

The first selection of articles was based on the keywords: new coronavirus infection, SARS-CoV-2, COVID-19, severe course, complications, thrombosis, cytokine storm, ACE-2, biomarkers. A total of 3714 publications was selected using the keywords, eight of them corresponding to all the inclusion criteria (Figure 1).

The inclusion criteria for original articles were publications describing the study design, the number of patients with COVID-19 infection, detection of immunogenetic factors and ACE-2 expression. Reviews, recommendations, results of clinical trials, publications dedicated to the analysis of functional and immunological disorders in COVID-19 patients and common descriptions of clinical symptoms among sick children (0 to 18 years old) and adults (after 18 years old) without division were out.

According to the analysis, eight studies with a description of various genetic factors that determine the severity of COVID-19 have been selected (Table 1).

## 3. Pathogenesis of the New Coronavirus Infection

According to the latest data on the pathogenesis of the new coronavirus infection, the key factor in the penetration of the SARS-CoV-2 virus into the cell is the expression of the ACE-2 protein, which explains the multiple organ damage in the disease. SARS-CoV-2 can enter cells expressing ACE-2, but not cells without ACE-2 or cells expressing other coronavirus receptors such as aminopeptidase N and dipeptidyl peptidase 4 (DPP4), suggesting that ACE-2 is a cellular receptor for SARS-CoV-2 [15].

ACE-2 is expressed in almost all human organs more or less [16]. In the respiratory system, the traditional immunohistochemistry and the recently presented analysis of single-cell RNA sequencing showed that ACE-2 is mainly expressed on alveolar type II cells but is poorly expressed on the surface of epithelial cells of the oral mucosa, nasal cavity and nasopharynx. The different expression suggests that the lungs are the main target of SARS-CoV-2 [17,18]. Moreover, ACE-2 is highly expressed on myocardial cells, cells of the proximal renal tubules and urothelial cells and in the urinary bladder [19]. Biliary ducts and tissue damage have also been found in skeletal muscles and the central nervous system, as well as in the adrenal glands and thyroid gland [20,21].

Impairment of the anti-viral immune response mechanisms has a crucial role in the course of infection. SARS-CoV-2 is able to suppress the innate immune response mechanisms [22]. Studies on atypical pneumonia show that a multiple viral structural and non-structural proteins interfere with the action of interferon at various stages of signaling pathways, including by preventing viral RNA recognition [22,23], signaling through TBK1, an inhibitor of nuclear factor kappa B kinase subunit (IKKε), TRAF3, and IRF3 [24], which disrupts a signal transduction of STAT1, as well as by inducing host mRNA degradation and inhibiting protein translation [25]. The suppression of proinflammatory mechanisms promotes viral replication and leads to an increased pyroptosis—an inflammatory form of programmed cell death that is usually observed in cytopathic viral infections [26]. There is an increased release of pyroptosis products, which can further induce aberrant inflammatory reactions [27,28]. The appearance of dead viral cell antigens stimulates the immune system to produce a large number of pro-inflammatory factors. The described processes cause activation of alveolar macrophages, lectin pathway of complement cascade and the formation of local immune complexes that enhance pro-inflammatory processes. The complement activation leads to endothelial damage and also induces leukocytes of the C3a and C5a components to produce proinflammatory cytokines: Interleukin IL-1, IL-6, IL-8, and IFN-γ [29]. The concentration of IL-6 produced by monocytes, macrophages and dendritic cells increases significantly [30]. IL-6 binds to the IL-6 receptor expressed mainly on immune cells, which is the cis-mechanism and, according to the trans-mechanism, to the soluble IL-6 receptor, forming a complex that can affect all cells that do not express membrane IL-6 receptor, including endothelial cells [31]. The massive activation causes a “cytokine storm” characterized by the production of vascular growth factor (VEGF), monocyte chemoattractant protein-1 (MCP-1), IL-8 and additionally IL-6. Secondary hemophagocytic lymphohistiocytosis (macrophage activation syndrome), a hyperinflammatory syndrome, characterized by the release of cytokines, cytopenia and multi-organ failure, develops. In addition to high levels of cytokines in patients, there is an increase in ferritin levels [32,33]. A decrease in the E-cadherin expression on endothelial cells contributes to an increase in vascular permeability, which leads to a decrease in blood pressure and pulmonary dysfunction [29]. Possible pathogenetic changes in the immune response in COVID-19 are presented on Figure 2.

We propose two groups of possible biomarkers to predict the severe course of new coronavirus infection—markers associated with the immune response characteristics and with the expression of ACE-2 system proteins.

## 4. Possible Genetic Predictors of Severe COVID-19

### 4.1. Genetic Characteristics of the Immune Response in Patients with Severe COVID-19

One of the most studied risk factors that predetermines the development of autoimmune inflammation is the Human leukocyte antigens (HLA) gene polymorphism [34]. These genes encode major histocompatibility complex (MHC) molecules that present antigens on the cell surface to T-lymphocytes. The HLA system is one of the first to come into contact with foreign antigens, which explains its effect on the development of the subsequent immune response, especially on autoimmune processes [35].

One of the earliest mentions of the role of HLA genes in the infection is a small study in the Han population, which showed that the HLA-C*07:29 and B*15:27 alleles may be associated with susceptibility to SARS-CoV-2 [36]. According to an in silico analysis, coronavirus proteins bind to MHC-B*46:01 molecules with the lowest binding affinity, which is why carriers of HLA-B*46:01 genotype may be especially vulnerable to COVID-19, as was previously shown for SARS-CoV-2 [37]. However MHC-A*02:02, -B*15:03, -C*12:03 molecules showed the greatest ability to present highly conserved SARS-CoV-2 peptides shared by human coronaviruses, suggesting that it may provide cross-protection mediated by T-cell immunity [37]. Also, in this study other MHC molecules have been found with low affinity for coronavirus proteins: MHC-A*25:01, C*01:02.

Genome-wide analysis of patients from Spain and Italy revealed a relationship between the severe course of the disease and the features of the 3p21.31 locus, which included six genes: SLC6A20, LZTFL1, CCR9, FYCO1, CXCR6, and XCR1 [8]. In the examined subjects were found decreased expression of the CXCR6 gene encoding chemokine receptor 6 with the CXC motif (CXCR6), which regulates the specific location of lung tissue-resident-memory CD8+ T cells, specific against respiratory system pathogens, including influenza viruses [38]. Also, the clinical significance was shown for increased expression of SLC6A20, which encodes sodium/imino-acid (proline) transporter 1 (SIT1) and which functionally interacts with angiotension-converting enzyme 2, the cell surface receptor SARS-CoV-2 [39]. Also researchers performed an analysis of HLA-genotyping of seven HLA loci (HLA-A, -C, -B, -DRB1, -DQA1, -DQB1, -DPB1) and correlation with COVID-19 severity (oxygen supplementation or mechanical ventilation). They did not find an association between the predisposition to severe COVID-19 and the heterogeneity of the HLA-alleles. Among the genes studied, CCR9 can be distinguished, encoding a homeostatic and inflammatory chemokine receptor. It is acknowledged that this receptor is involved in the pathogenesis of pneumonia of various origins and its expression is induced in the early stages of airway inflammation. CCR9 and its ligand CCL25 have been shown to play an important role in modulating the recruitment of eosinophils and lymphocytes in the early stages of inflammation [40].

Lu et al., in 2020, found several genes whose mutations were associated with an increased risk of death in COVID-19. When studying the genotype of 193 infected patients, scientists identified four genes with missense variants—the rs150892504 allele of the ERAP2 gene, the s138763430 allele of the BRF2 gene, the rs117665206 allele of the TMEM181 gene, the rs147149459 allele of the ALOXE3 gene. Among the described mutations, it is possible to distinguish alterations in the ERAP2 gene encoding metalloaminopeptidase involved in the final modification of antigens for presentation by MHC-I molecules [9].

A small study of four clinical cases of severe coronavirus infection was carried out, which found, using whole-genome sequencing, loss-of-function variants in X-chromosomal TLR7 gene encoding the toll-like receptor 7 in patients and their relatives. A decrease in the type I interferon (IFN) signaling was revealed in peripheral blood mononuclear cells from patients, which was expressed as a decrease in the mRNA expression of proteins IRF7, IFNB1 and ISG15 upon stimulation with imiquimod, a TLR7 agonist, as well as decreased production of IFN-γ, IFN type II [10]. According to fundamental studies, the X chromosomal TLR7 gene is associated with the maintenance of the innate immune response against coronaviruses [41,42].

According to Zhang et al.’s 2020 analysis, homozygous inheritance of the rs12252 allele of the interferon-induced transmembrane protein 3 (IFITM3) may be associated with severe disease [14]. IFITM3 encodes an immune effector protein that enhances the accumulation of CD8+ T cells in airways to promote mucosal immune cell persistence, which is critical in viral infections [43,44]. This protein is considered to participate in viral invasion and be associated with COVID-19 severity and cytokine release syndrome [45].

### 4.2. Specificity of ACE-2 Protein Expression in Patients with Severe COVID-19

The ACE protein plays a key role in the virus entry into cells. Endocytosis of virions into the cell is initiated by the receptor-binding domain (RBD) of the S1 subunit of the surface S (spike) protein [46,47]. Inside the endosome, the S1 subunit is cleaved, due to which the fusion protein of the S2 subunit is inserted into the endosome membrane and the virus is released into the cell cytoplasm [48,49].

Despite the fact that the receptor-binding domain (RBD) of SARS-CoV and SARS-CoV-2 has a 72% amino acid sequence similarity, the RBD protein of SARS-CoV-2 binds to ACE-2 with a higher affinity than SARS-CoV [50,51]. This may be related to significant differences in the RBD of SARS-CoV-2 and SARS-CoV. In particular, of the 33 amino acids in this region (amino acids 460-492) in the SARS-CoV S protein, which contains critical residues and binds to ACE-2 52, less than half of them (15/33) are conservative in SARS-CoV-2 [52]. However, murine anti-SARS-CoV protein antiserum can cross-neutralize the SARS-CoV-2 pseudovirus, suggesting overlapping neutralizing epitopes between the two viruses [53].

Moreover, the SARS-CoV-2 S protein contains a furin-like cleavage site, as in the MERS-CoV virus and the human coronavirus HCoV, which is not observed in SARS-CoV 53. These characteristics may contribute to the increased infectivity of SARS-CoV-2 over SARS-CoV. In addition to the preliminary cleavage by furin, the transmembrane protease serine subtype 2 is also required for the correct processing of the SARS-CoV-2 spike protein and facilitating entry into the host cells [54].

Thus, the importance of studying the structure and function of ACE-2 in the human body can be emphasized. ACE is believed to have a protective function against lung damage, therefore damage of cells expressing ACE2 leads to a decrease of these molecules and the protective function of ACE-2 [55]. Overexpression of human ACE-2 has been shown to increase the severity of the disease in a mouse model of SARS-CoV infection, demonstrating that viral cell penetration is critical [56,57] and injection of SARS-CoV spike protein in mice exacerbated lung damage [58]. According to the analyzed data, SARS-CoV not only uses ACE as a carrier into the cell but also, by disrupting the ACE function, reduces the reparative capacity of the lung tissue [59].

The results of the various studies described above indicate the association between the degree of expression and the types of configuration of ACE-2 and related proteins in susceptibility to SARS-CoV-2 and predisposition to severe disease [59].

As already described above, association between the overexpression of the SLC6A20 gene located at the 3p21.31 locus and encoding the transport protein for ACE-2 with the more severe course of COVID-19 was found [8].

According to the degree of expression of ACE-2 and TMPRSS2 transport proteins, the researchers developed a treatment algorithm for patients [11]. Patients with wild-type or naïve ACE-2 and TMPRSS2 expression have a severe form, which requires combination treatment, in the presence of TMPRSS2 polymorphism or dysregulation, monotherapy with antiviral drugs is sufficient, and the disease is mild with ACE-2 polymorphism or dysregulation. At the same time, the researchers identified the racial features in the expression of various ACE and TMPRSS2 alleles.

There have been various clinical studies of ACE expression. A recent analysis of a single-cell RNA-seq showed that Asian males may have a higher expression of ACE-2 than women [12].

According to a study by Cao et al., there were no naturally resistant mutations for coronavirus S-protein binding in the Chinese population [13]. Interestingly, SNP rs2285666 with the highest frequency in 62 variants showed a much higher frequency in the population of China-MAP (0.556) and CHS (Han, Southern China, 0.557) compared with other populations (Ad Mixed American (AMR) 0.336; Africans (AFR) 0.2114; European (Euro) 0.235). Moreover, in the Chinese population, the frequency of homozygous mutations was significantly higher among men (0.550) than among women (0.310). Their results showed that the genotypes of the ACE-2 gene polymorphism can be associated with higher levels of ACE-2 expression in the East Asian population.

Based on the analysis results, it is possible to suggest possible biomarkers that predict the development of severe COVID-19, which can be determined using whole blood cell sequencing (Table 2).

At the moment, it is difficult to argue about the diagnostic significance of the described genetic factors. To determine their prognostic effectiveness, large-scale population studies are required, taking into account many additional factors that can worsen the course of the disease (age, concomitant diseases, etc.). It should also be noted that genetic analysis is expensive and its use will be limited, which will not allow it to be used as a screening test.

### 4.3. Gender-Related Specifics of COVID-19 

The mortality rate difference between men (2.8%) and women was identified (1.7%) [59] There are gender differences in the described possible genetic factors of severe COVID19, which may explain the difference in the number of severe and fatal cases.

It was shown that the mean serum concentrations of SARS-CoV-2 IgG antibodies were higher among women than among men at an early stage of infection, as well as with the development of severe COVID-19 [60,61]. Such differences can be explained by different immunoregulatory functions of estrogen and testosterone [62,63]. By activating the estrogen receptor, estrogens regulate the development of immune cells and the pathways of the immune system response against infections (and autoantigens), which leads to B-cell-mediated adaptive immunity and the production of specific antibodies [64]. During the innate immune response, estrogen also, of the activation of its receptor on monocytes, macrophages and neutrophils, induces the production of pro-inflammatory cytokines (IL-12, TNF-α) and chemokines (CCL2). These pro-inflammatory mediators activate lymphocytes and alveolar macrophages with increased production of type I and III interferon (IFN), which is important for reducing the production of viral titers [65]. Conversely, testosterone has an immunosuppressive effect and does not significantly affect inflammatory and reparative immune functions [66].

Moreover, there are gender-related differences in ACE-2 expression. Sex hormones in men might cause a higher expression of ACE-2 compared with women, with a higher frequency of infection [12]. Gender differences are also explained by the fact that the ACE-2 gene is located on the X chromosome, which means that there may be alleles that confer resistance to COVID-19, which explains the lower mortality rate among women. Polymorphisms in the TMPRSS2 gene, including p.Val160Met (rs12329760), associated with a genetic predisposition to COVID-19, were found to be more specific for male patients [11]. Higher expression of ACE-2 gene [12] and the presence of homozygous mutations [13] among men have been found in other studies.

### 4.4. Specifics of the COVID-19 Course in Various Populations 

Connection analyses between genetic markers and severe disease in populations show conflicting results. On the one hand, HLA genotypes of predisposition to the development of severe infection were identified, which was even associated with the travels of Marco Polo [67]. Analyzing maps of his movements and geographical comparison maps HLA (www.igdawg.org/software/browser-beta.html) (accessed on 11 November 2020), researchers hypothesized the presence of additional HLA alleles, such as HLA-A*02:01, HLA-DPA1*01:03, HLA-B*07:04, HLA-B*13:02, HLA-B*14:03, HLA-C*01:01 и HLA-C*05:01, which are common for Italian and Chinese populations [67,68]. Nguyen et al. (2020), according to in silico models, built maps of the distribution of HLA-A*02:02, B*15:03, C*12:0 genotypes presumably associated with low susceptibility to coronaviruses, and HLA-A*25:01, B*46:01, C*01:02 genotypes presumably associated with a predisposition to coronavirus infection, among various populations [37] (Table 3).

Various studies have indicated polymorphisms and mutations of the ACE-2 gene associated with a more severe disease and occurring mainly in the Chinese population [55].

According to the study by Hou et al. [11], it is possible to distinguish the race features of the inheritance of genes associated with ACE system proteins. Specifically, 39% (24/61) and 54% (33/61) of the predisposing ACE-2 variants occur in African/African American (AFR) and non-Finnish European (EUR) populations, respectively. The prevalence of predisposing gene variants among Hispanic/Mixed American (AMR), East Asian (EAS), Finnish (FIN) and South Asian (SAS) populations is 2–10%, while such gene variants in the ACE-2 encoding regions among Amish (AMI) and Ashkenazi (ASJ) were not found. At the same time, polymorphisms of the ACE-2 and TMPRSS2 genes are more common among Africans and Europeans, facilitating easier penetration of the virus into the cell, which may explain the differential genetic predisposition to COVID-19 [11].

Table 2 presents comparative data of the most characteristic gene markers and incidence rate among different populations.

According to the comparative analysis statistical data on the frequency of severe cases, the highest incidence of COVID-29 cases was recorded in Europe and Asia, with the level being the lowest in Africa. Perhaps the HLA genotypes reflecting the affinity properties of MHC molecules to virus proteins can really serve as an explanation for this pattern. Thus, genotypes with low affinity to the virus are most often found among the East Asian population, while those with high affinity are found in the African population. However, it is worth considering the limitations associated with the method, statistical data processing, the level of health care and literacy of the population in each country.

## 5. Conclusions

According to the analysis of the literature describing the association of immunogenic characteristics and the expression of ACE-2 and related proteins with the development of severe COVID-19, we can propose the following genetic factors as biomarkers-predictors—genes, associated with immune response (HLA genotype, genes coding cytokines and proinflammatory mediators receptors) and specifics in ACE-2 expression, and its transporter system proteins (SLC6A20, TMPRSS2 genes). The review showed that severe biomarkers are more common for men. Today, the presence of population differences in COVID-19 course can only be assumed.

This study is a review of some experimental and clinical data therefore it cannot give strong recommendations to use biomarkers described in clinical practice. However, it is worth noting that the determination of the described markers at the early stages of the disease may help identify high-risk groups that require monitoring for COVID-19 infection and an alternative algorithm for treating the infection at the earliest stages of its development, which will allow reduction of the risk of severe cases in the population and will contribute to the development of epidemiological measures specific for each region. Therefore, the search for biomarkers for the prediction of severe COVID 19 has great medical, economical and epidemiological importance for every country.

## Figures and Tables

**Figure 1 vaccines-09-00211-f001:**
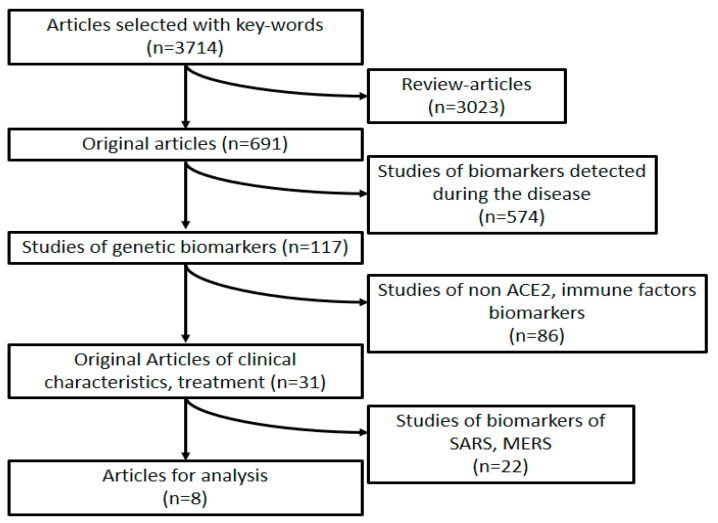
Design of the study.

**Figure 2 vaccines-09-00211-f002:**
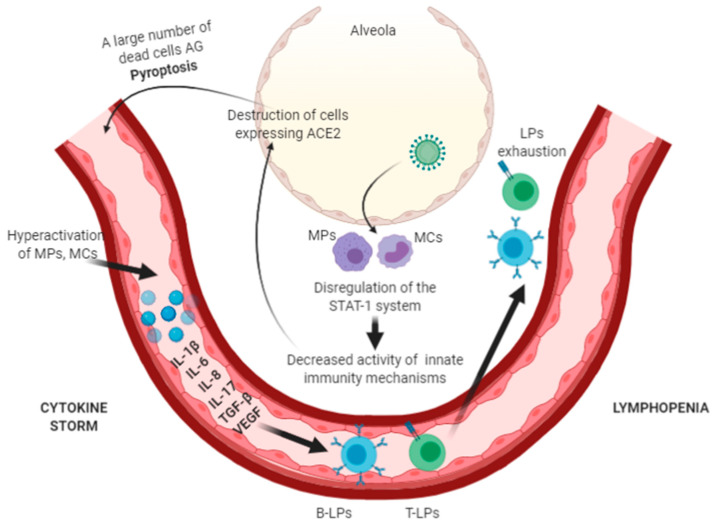
Possible pathogenetic changes in the immune response in COVID-19.

**Table 1 vaccines-09-00211-t001:** Studies Selected for Analysis.

Author	Country	Number of Patients	Study Variables	Method
Wang et al., 2020 [7]	China	82	HLA	Next-generation sequencing (NGS)
The Severe Covid-19 genome-wide association study (GWAS) Group, 2020 [8]	Spain Italy	1980	Single-nucleotide polymorphisms	Global Screening Array (GSA)
Lu et al., 2020 [9]	UK	193 deaths from 1412 confirmed infections in the 5871 group	Distribution of genetic variants	GWAS
Van Der Made et al., 2020 [10]	The Netherlands	4	Distribution of genetic variants	Rapid clinical whole-exome sequencing
Hou et al., 2020 [11]	China	81,000 human genomes	Distribution of genetic variants	Work with databases
Zhao et al., 2020 [12]	China	8	Single-cell RNA-sequencing	Unsupervised graph-based clustering
Cao et al., 2020 [13]	China	1700 variants	Functional coding variants in ACE-2	Work with databases
Zhang et al., 2020 [14]	China	80	300 bp locus spanning rs12252	Next-generation sequencing (NGS)

**Table 2 vaccines-09-00211-t002:** Genetic factors of severe COVID-19 course.

Marker Type	Marker	Pathogenetic Effect
Associated with the immune response	HLA-B*46:01/A25:01/C01:02	Suitable MHC molecules have a low affinity for the virus proteins, which can make the representatives of this genotype especially vulnerable
	Hypoexpression of the CXCR6 gene	Encodes chemokine receptor 6 with the CXC motif (CXCR6), which regulates the specific location of lung tissue-resident-memory CD8+ T cells specific against respiratory pathogens
	Polymorphism of the CCR9 gene	Encodes a homeostatic and inflammatory chemokine receptor, which is involved in the pathogenesis of pneumonia of various origins and its expression is induced in the early stages of airway inflammation.
	X-chromosomal TLR7 gene	Encodes toll-like receptor 7, is associated with the maintenance of the innate immune response against coronaviruses, with the IFN-γ production
	rs150892504 mutation of the ERAP2 gene	Encodes metalloaminopeptidase involved in the final modification of antigens for presentation by MHC-I molecules
	rs12252 allele of IFITM3 gene	Encodes interferon-induced transmembrane protein 3, which enhances the accumulation of CD8+ T cells in airways to promote mucosal immune cell persistence
ACE-2 protein-associated	ACE-2 wild-type or naïve expression	Risk of severe disease
	ACE-2 allele P.Arg514Gly	Affects the renin-angiotensin system (RAS) function—the risk of cardiovascular and pulmonary complications
	ACE-2 alleles p.Arg708Trp p.Arg710Cys p.Arg710His, p.Arg716Cys	Facilitates the penetration of SARS-S into host cells
	Overexpression of the SLC6A20 gene	Encodes sodium/imino-acid (proline) transporter 1 (SIT1), which functionally interacts with angiotension-converting enzyme 2
	TMPRSS2 wild-type or naïve expression	Risk of severe disease
	TMPRSS2 p.Val160Met allele	Risk of severe disease among men

**Table 3 vaccines-09-00211-t003:** Gene markers and data of the incidence rate COVID-19 during December 2020.

Population	HLA Genotypes with a Possible Protective Function	Predisposing HLA Genotypes	ACE Expression	Number of Cases	Number of Severe Cases	% of Severe Cases
Africans/ African American (AFR)	C*12:03 A*02:02 B*15:03		polymorphisms of differential genetic susceptibility to COVID-19	2,199,524	2.611	0.12%
Non-Finnish Europeans (EUR)	C*12:03	A*25:01	polymorphisms of differential genetic susceptibility to COVID-19	17,480,113	28.587	0.16%
Hispanic/Mixed-race American (AMR)	A*02:02 C*12:03	C*01:02		11,216,585	17.025	0.15%
East Asians (EAS)	C*12:03	A*25:01 B*46:01 C*01:02		16,893,952	27.397	0.16%

## Data Availability

Data available in a publicly accessible repository.

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
