# Peer review of "Immunogenetic Predictors of Severe COVID-19"

_vaccines, 2021, doi:10.3390/vaccines9030211_

Round 1

Reviewer 1 Report

Line 196 is unclear. Is this a typo? Some abbreviations in the middle of the article ?

Abbreviations: ACE-2 – angiotension-converting enzyme 2, AG – antigens, IL – in- terleukins, LP – lymphocytes, MPs – macrophages, MCs – monocytes, TGF-β - Transform- ing growth factor beta, VEGF- vascular endothelial growth factor.

Authors mentioned, “Considering genetic factors, several publications on the predisposition to the development of coronavirus infection can be found today. One of the most studied risk factors 148 that predetermine the development of autoimmune inflammation is the HLA gene poly-149 morphism”

Could be better to show a Table with these genetic factors.

Line 176: “At the same time, this study did not find an association between 173 the predisposition to severe COVID-19 and the heterogeneity of the HLA loci”

Before this line, there are no results. How the authors concluded this? Explain.

Line 197: Interferon has been related to severity. Is this studied in this article? Please, add some comment in line 197.

Line 242 “There have been various clinical studies of ACE expression. A recent analysis of a  single-cell RNA-seq showed that Asian males may have a higher expression of ACE-2[12].”

Higher than ?

Line 222. ACE has a protective function against lung damage. It seems that is good then. Comment on the paper.

Line 279. Higher expression (?) [12] . Please write this better.

Line 309. “According to the comparative analysis statistical data on the frequency of severe 309 cases, the highest incidence of COVID-29 cases was recorded in Europe and Asia, with the  level being the lowest in Africa. Perhaps, the HLA genotypes reflecting the affinity properties of MHC molecules to virus proteins can really serve as an explanation for this pattern. Thus, genotypes with low affinity to the virus are most often found among the East  Asian population, while those with high affinity are found in the African population.”

Table 3 mention the biomarkers.

Please add comments and references about type of blood affecting the severity.

The authors need to expand subsection 3.5. Is too short without important content. Talk more about other predictors and uncertainty about these predictors. Confounding factors? Please, comment on limitations of finding predictors.

Please rewrite lines 331-332-333. It is not clear what the authors want to say.

Conclusions should be extend in relationship to 3.5. Moreover, emphasize that is a review article.

Author Response

Dear reviewer,

Thank you very much for such deep and careful analysis of the article!

We are very grateful for all your valuable comments, what undoubtedly will improve our work!

Line 196 is unclear. Is this a typo?

205: IFN-γ - IFN type II

Some abbreviations in the middle of the article ?

Abbreviations: ACE-2 – angiotension-converting enzyme 2, AG – antigens, IL – in- terleukins, LP – lymphocytes, MPs – macrophages, MCs – monocytes, TGF-β - Transform- ing growth factor beta, VEGF- vascular endothelial growth factor.

were transferred before abstract

Authors mentioned, “Considering genetic factors, several publications on the predisposition to the development of coronavirus infection can be found today. One of the most studied risk factors 148 that predetermine the development of autoimmune inflammation is the HLA gene poly-149 morphism” Could be better to show a Table with these genetic factors.

156: The first sentence was removed

The table 2 contains all analyzed genetic factors

Line 176: “At the same time, this study did not find an association between 173 the predisposition to severe COVID-19 and the heterogeneity of the HLA loci” Before this line, there are no results. How the authors concluded this? Explain.

182: Comment: It was the highline from the article.

Now is corrected: Also researchers performed analysis of HLA-genotyping of seven HLA loci (HLA-A, -C, -B, -DRB1, -DQA1, -DQB1, -DPB1) and correlation with Covid-19 severity (oxygen supple-mentation or mechanical ventilation). They did not find an association between the pre-disposition to severe COVID-19 and the heterogeneity of the HLA-alleles.

Line 197: Interferon has been related to severity. Is this studied in this article? Please, add some comment in line 197.

208: Was corrected:

According to Zhang et al., 2020. analysis, homozygous inheritance of the rs12252 allele of the interferon-induced transmembrane protein 3 (IFITM3) may be associated with severe disease [14]. IFITM3 encodes an immune effector protein, that enhances the accumulation of CD8+ T cells in airways to promote mucosal immune cell persistence, what is critical in viral infections [43,44]. This protein is considered to participate in viral invasion and be associated with COVID-19 severity and cytokine release syndrome [45].

Line 242 “There have been various clinical studies of ACE expression. A recent analysis of a  single-cell RNA-seq showed that Asian males may have a higher expression of ACE-2[12].”Higher than ?

Was added: than women

Line 222. ACE has a protective function against lung damage. It seems that is good then. Comment on the paper. – there is double pathogenic effect: the more ACE2

235: Was edited:

ACE is believed to have a protective function against lung damage, therefore damage of cells expressing ACE2 leads to decrease of these molecules and protective function of ACE-2 

Line 279. Higher expression (?) [12] . Please write this better.

258: Was added: of ACE-2 gene

Line 309. “According to the comparative analysis statistical data on the frequency of severe 309 cases, the highest incidence of COVID-29 cases was recorded in Europe and Asia, with the  level being the lowest in Africa. Perhaps, the HLA genotypes reflecting the affinity properties of MHC molecules to virus proteins can really serve as an explanation for this pattern. Thus, genotypes with low affinity to the virus are most often found among the East  Asian population, while those with high affinity are found in the African population.” Table 3 mention the biomarkers.

The term “Biomarker” was exchanged on genetic/immunogenetic factors

Please add comments and references about type of blood affecting the severity.

According to study design and the second review`s suggestions to focus only on the relevant information the study is focused only on analysis of genes, associated with immune response and ACE2 expression

The authors need to expand subsection 3.5. Is too short without important content. Talk more about other predictors and uncertainty about these predictors. Confounding factors?

According to the second review`s suggestions the structure of the article was changed and subsection 3.5 was transferred in the end of 3.2 as a summarization of results obtained

Please, comment on limitations of finding predictors.

273: Was added: At the moment, it is difficult to argue about the diagnostic significance of the de-scribed genetic biomarkers. To determine their prognostic effectiveness, large-scale population studies are required, taking into account many additional factors that can worsen the course of the disease (age, concomitant diseases, etc.). It should also be noted that genetic analysis is expensive, and its use will be limited, which will not allow it to be used as a screening test.

Please rewrite lines 331-332-333. It is not clear what the authors want to say.

The conclusion was edited:

According to the analysis of the association of immunogenic characteristics and the expression of ACE-2 and related proteins with the development of severe COVID-19 showed that the following biomarkers were identified: genes, associated with immune response (HLA genotype, genes coding cytokines and proinflammatory mediators receptors), and specifics in ACE-2 expression, and it`s transporter system proteins (SLC6A20, TMPRSS2 genes). According to the analysis data, severe biomarkers are more common for men. Today, the presence of population differences in COVID-19 course can only be assumed.

The study is a review of some experimental and clinical data therefore it can`t give strong recommendations to use biomarkers described in clinical practice. However, its worthy to note that determination of the described markers at the early stages of the disease may help identify high-risk groups that require monitoring for COVID-19 infection and an alternative algorithm for treating the infection at the earliest stages of its development, which will allow reducing the risk of severe cases in the population and will contribute to the development of epidemiological measures specific for each region. Therefore the search of biomarkers for prediction of severe COVID 19 has a great medical, economical and epidemiological importance for every country.

Conclusions should be extend in relationship to 3.5. Moreover, emphasize that is a review article.

The conclusion was edited:

According to the analysis of the association of immunogenic characteristics and the expression of ACE-2 and related proteins with the development of severe COVID-19 showed that the following biomarkers were identified: genes, associated with immune response (HLA genotype, genes coding cytokines and proinflammatory mediators receptors), and specifics in ACE-2 expression, and it`s transporter system proteins (SLC6A20, TMPRSS2 genes). According to the analysis data, severe biomarkers are more common for men. Today, the presence of population differences in COVID-19 course can only be assumed.

The study is a review of some experimental and clinical data therefore it can`t give strong recommendations to use biomarkers described in clinical practice. However, its worthy to note that determination of the described markers at the early stages of the disease may help identify high-risk groups that require monitoring for COVID-19 infection and an alternative algorithm for treating the infection at the earliest stages of its development, which will allow reducing the risk of severe cases in the population and will contribute to the development of epidemiological measures specific for each region. Therefore the search of biomarkers for prediction of severe COVID 19 has a great medical, economical and epidemiological importance for every country.

Reviewer 2 Report

Malkova et al., performed a literature review to identify the genetic biomarkers characterizing the immune response in the severe SARS-CoV-2 infection. However, this paper’s objective is timely; the representation of this paper needs to be modified for publication. The following major issues should be considered in the revision of the manuscript to reconsider it for publication.

Specific issues:

  1. This paper is not in the standard format of a review paper. The authors used different sections like methods, results and discussion, conclusion, which is the format of a general research article. So, I’ll suggest rewriting the manuscript to make it an interesting review paper.
  2. The manuscript should be substantially shortened and focus only on the relevant information.
  3. On page 3, line 87-88, authors stated that “The meta-analysis was carried out in accordance with the PRISMA protocol”. But they did not provide any statistics related to the meta-analysis. If meta-analysis is performed, they should provide some of the important statistics related to meta-analysis, such as odds ratio. A test of heterogeneity should be performed to check if there is any heterogeneity among the literatures that they selected for their studies. Moreover, the author should add some statistics and plot to show that the different literatures included in their study are significant.
  4. The authors should finish the manuscript with a strong take-home message - not just by briefly summarizing the state of current knowledge but also by indicating what is likely to be the most productive avenues for future research and by highlighting current and future limitations.

Author Response

Dear reviewer,

Thank you very much for such deep and careful analysis of the article!

We are very grateful for all your valuable comments, what undoubtedly will improve our work!

“Malkova et al., performed a literature review to identify the genetic biomarkers characterizing the immune response in the severe SARS-CoV-2 infection. However, this paper’s objective is timely; the representation of this paper needs to be modified for publication. The following major issues should be considered in the revision of the manuscript to reconsider it for publication.”

This paper is not in the standard format of a review paper. The authors used different sections like methods, results and discussion, conclusion, which is the format of a general research article. So, I’ll suggest rewriting the manuscript to make it an interesting review paper.

The structure was edited:

1.Introduction

2. Design of the study

3. Pathogenesis of the new coronavirus infection

4. Possible genetic predictors of severe COVID-19

4.1. Genetic characteristics of the immune response in patients with severe COVID-19

4.2. Specificity of ACE-2 protein expression in patients with severe COVID-19

4.3. Gender-related specifics of COVID-19

4.4. Specifics of the COVID-19 course in various populations

5. Conclusions

The manuscript should be substantially shortened and focus only on the relevant information.

I this study we tried to deeply research pathogenesis of the Coronovirus infection to reveal pathogenic aspects, that can reflect the severity of the disease and can be used as biomarkers. After that we described the results of researches, confirming our hypothesis.

On page 3, line 87-88, authors stated that “The meta-analysis was carried out in accordance with the PRISMA protocol”. But they did not provide any statistics related to the meta-analysis. If meta-analysis is performed, they should provide some of the important statistics related to meta-analysis, such as odds ratio. A test of heterogeneity should be performed to check if there is any heterogeneity among the literatures that they selected for their studies. Moreover, the author should add some statistics and plot to show that the different literatures included in their study are significant.

99

The meta-analysis wasn`t performed

The part about statistical analysis was removed

The authors should finish the manuscript with a strong take-home message - not just by briefly summarizing the state of current knowledge but also by indicating what is likely to be the most productive avenues for future research and by highlighting current and future limitations.

The conclusion was edited:

According to the analysis of the association of immunogenic characteristics and the expression of ACE-2 and related proteins with the development of severe COVID-19 showed that the following biomarkers were identified: genes, associated with immune response (HLA genotype, genes coding cytokines and proinflammatory mediators receptors), and specifics in ACE-2 expression, and it`s transporter system proteins (SLC6A20, TMPRSS2 genes). According to the analysis data, severe biomarkers are more common for men. Today, the presence of population differences in COVID-19 course can only be assumed.

The study is a review of some experimental and clinical data therefore it cant give strong recommendations to use biomarkers described in clinical practice. However, its worthy to note that determination of the described markers at the early stages of the disease may help identify high-risk groups that require monitoring for COVID-19 infection and an alternative algorithm for treating the infection at the earliest stages of its development, which will allow reducing the risk of severe cases in the population and will contribute to the development of epidemiological measures specific for each region. Therefore the search of biomarkers for prediction of severe COVID 19 has a great medical, economical and epidemiological importance for every country.

Round 2

Reviewer 1 Report

Line 25:  The authors mentioned that that the following genetic factors were significant. This is a very strong conclusion that is not supported by this study. There are many factors and confounders, that are not analyzed or are not conclusive. Please, emphasize the uncertainty in the correlation.  This line should be changed to mention something like the review suggest that….

Notice that in the conclusions it says

“273: Was added: At the moment, it is difficult to argue about the diagnostic significance of the described genetic biomarkers. To determine their prognostic effectiveness, large-scale population studies are required, taking into account many additional factors that can worsen the course of the disease (age, concomitant diseases, etc.). It should also be noted that genetic analysis is expensive, and its use will be limited, which will not allow it to be used as a screening test.”

Line 77. The word hypotheses should be changed to emphasize that there is not statistical study in the paper. Maybe something like review exploration.

Line 343. The phrase sounds conclusive, but it should be just exploratory results. In addition, the phrase is not clear. Please, mention that you review some literature and found that ….

Line 351. This is study is a review .

Author Response

Dear reviewer,

Thank you very much for your comments!

With your help the article was notably improved, it`s a great honor to work with such expert in this field

Line 25:  The authors mentioned that that the following genetic factors were significant. This is a very strong conclusion that is not supported by this study. There are many factors and confounders, that are not analyzed or are not conclusive. Please, emphasize the uncertainty in the correlation.  This line should be changed to mention something like the review suggest that….

Notice that in the conclusions it says

“273: Was added: At the moment, it is difficult to argue about the diagnostic significance of the described genetic biomarkers. To determine their prognostic effectiveness, large-scale population studies are required, taking into account many additional factors that can worsen the course of the disease (age, concomitant diseases, etc.). It should also be noted that genetic analysis is expensive, and its use will be limited, which will not allow it to be used as a screening test.”

The answer:

Line 35: was edited:

The literature analysis of the association of immunogenic characteristics and the expression of ACE-2 and related proteins with the development of severe COVID-19 revealed following genetic factors: HLA-B*46:01 genotype, CXCR6 gene hypoexpression, CCR9 gene expression, TLR7, rs150892504 mutations in the ERAP2 gene, overexpression of wild-type ACE-2, TMPRSS2 and its` different polymorphisms. Genes, associated with the severe course, are more common among men. Ac-cording to the analysis data, it can be assumed that there are population differences. However, diagnostical significance of the markers described must be confirmed with additional clinical studies. 

Line 77. The word hypotheses should be changed to emphasize that there is not statistical study in the paper. Maybe something like review exploration.

The answer:

Line 77: hypotheses was exchanged on review exploration

Line 343. The phrase sounds conclusive, but it should be just exploratory results. In addition, the phrase is not clear. Please, mention that you review some literature and found that ….

The answer:

Was edited:

According to the analysis of literature describing the association of immunogenic characteristics and the expression of ACE-2 and related proteins with the development of severe COVID-19 we can propose the following genetic factors as biomarkers-predictors: genes, associated with immune response (HLA genotype, genes coding cytokines and proinflammatory mediators receptors), and specifics in ACE-2 expression, and it`s trans-porter system proteins (SLC6A20, TMPRSS2 genes). The review showed, that severe biomarkers are more common for men. Today, the presence of population differences in COVID-19 course can only be assumed. 

Line 351. This is study is a review .

The answer: was corrected

Reviewer 2 Report

The manuscript has been properly revised according to the suggestion.

Author Response

Dear review,

thank you very much for your help!!!